# EXPLOITING FINE-TUNING STRUCTURES TO IMPROVE ADVERSARIAL TRANSFERABILITY ON DOWNSTREAM SAM

## ABSTRACT

Combining the Segment Anything Model (SAM) with fine-tuning techniques allows SAM to be effectively adapted to various downstream image segmentation tasks. However, this adaptability introduces new security vulnerabilities related to adversarial attacks. In this paper, we investigate the adversarial transferability between the original SAM and its fine-tuned downstream models. Under limited knowledge conditions of the downstream models, we propose a novel structure-exploiting transferable attack (SETA) method. Our framework mimics the fine-tuning architecture and estimates the parameter distributions of the downstream models to improve the transferability of the generated adversarial samples. Experimental results demonstrate the efficacy of our proposed method in creating adversarial examples against various downstream fine-tuned SAM models.

## 1 INTRODUCTION

The Segment Anything Model (SAM) (Kirillov et al., 2023) is a foundational model for image segmentation trained on the SA-1B dataset, demonstrating strong generalization to unseen image segmentation tasks with user prompts. However, recent studies have shown limitations in its effectiveness in specific domain-oriented tasks, such as nuclei segmentation (Na et al., 2024; Chen et al., 2024b), polyp detection (Li et al., 2024b; Zhou et al., 2023), and camouflaged object detection (Chen et al., 2023a). To address such limitations, parameter-efficient fine-tuning (PEFT) techniques have been developed for fast adaptation of SAM to downstream tasks (Zhang et al., 2023). Methods like LoRA (Hu et al., 2022) and Adapter (Wu et al., 2025) have proven to be more efficient and achieve better generalization compared to full-parameter fine-tuning of foundational models. However, integrating SAM with these fine-tuning mechanisms brings new security concerns, particularly vulnerability to adversarial attacks.

Existing studies (Zhu et al., 2024; Akhtar et al., 2021) demonstrated that a neural network model is vulnerable to adversarial attacks that craft well-designed samples, making the neural network model unable to classify or segment correctly. Such adversarial attacks utilize either full knowledge of the model and the dataset, called white-box attacks (Liu et al., 2020a), or limited knowledge of the training dataset with no access to the victim model, called black-box attacks (Ge et al., 2023; Xia et al., 2024; Zhu et al., 2024). Specifically, enhancing the transferability of adversarial attacks is one of the approaches for black-box attacks (Li et al., 2020b; Ge et al., 2023; Qin et al., 2022), which designs adversarial samples on one accessible model that can also be used to deceive another non-accessible victim model. This transferability of adversarial attacks also exists in SAM and its various downstream models, as SAM is an open-source model that intrinsically reveals the vulnerability of its downstream models (Xia et al., 2024).

However, known approaches to improving attack transferability often rely on a surrogate model trained using a subset of the original training data. These methods typically overlook fine-tuning scenarios, in which the victim model has been adapted to a different, domain-specific dataset. In Xia et al. (2024), adversarial attack transferability was first explored in the context of fine-tuned SAM models; it proposed UMI-GRAT attacks, which incorporate Gaussian noise to bridge the output gap between SAM and its downstream variants.

In this paper, we propose a novel framework that leverages fine-tuning structures to improve the transferability of adversarial attacks on downstream SAM models. The main contributions of our work are as follows.

1. We investigate the transferability of adversarial attacks from SAM to its fine-tuned downstream variants, under the assumption that the adversary has no knowledge of the downstream task dataset.

2. We introduce a novel structure-exploiting transferable attack (SETA) method, where the structure here stands for the added fine-tuning layer in SAM. Our SETA method exploits structural characteristics to enhance the transferability of adversarial samples to downstream SAM.

3. We perform extensive experiments across various downstream SAMs and datasets to evaluate our approach compared to state-of-the-art (SOTA) transferability-enhancing attack methods. The experimental results show that our approach outperforms existing methods.

## 2 RELATED WORK

### 2.1 FINE-TUNING SAM

SAM is a powerful tool for zero-shot image segmentation, but it often requires fine-tuning to perform well in specific domains or tasks. Recent study Zhang et al. (2023) has explored efficient ways to adapt SAM without retraining the entire model. Wu et al. (2025) introduced Med-SAM, a fine-tuned version of SAM trained on a large-scale medical image dataset, which incorporates adapters into both the image encoder and mask decoder. In Zhang & Liu (2023), it proposed another fine-tuning approach using a low-rank adaptation (LoRA) strategy, modifying the prompt encoder and mask decoder components for supervised medical image segmentation tasks. In addition to medical image applications, SAM has also been adapted to other domains through similar fine-tuning methods (Na et al., 2024; Zhong et al., 2024; Pu et al., 2025).

However, even without knowledge of the downstream fine-tuning task, the original SAM can reveal general vulnerabilities that are transferable to downstream models. (Xia et al., 2024) is the first work to address this transferable attack problem by generating adversarial samples using SAM without accessing the downstream model. It proposed MUI-GRAT, which introduces gradient-based noise augmentation to mitigate the gradient mismatch between SAM and its fine-tuned variants. But, this approach relies on heuristic noise configurations, limiting its generalizability across diverse downstream models.

### 2.2 TRANSFERABLE ADVERSARIAL ATTACKS

The transferability of adversarial attacks refers to scenes in which an adversarial example that successfully misleads one model is also able to mislead another model. Existing approaches for enhancing attack transferability can be categorized into optimization-based and generation-based methods. Optimization-based methods (Dong et al., 2018; Liu et al., 2020a; Ge et al., 2023; Qin et al., 2022; Li et al., 2020a; 2023) rely on one or more surrogate models and iteratively generate adversarial perturbations using gradient information. In contrast, generation-based methods (Poursaeed et al., 2018; Xiao et al., 2018; Zhao et al., 2023; Chen et al., 2024a; 2023b) train a generative model to produce adversarial examples.

However, both types of approaches assume that the adversary has access to the training dataset or the task of the victim model. This assumption brings challenges in more realistic scenarios where the adversary does not have knowledge of the downstream task and dataset. Such situations are common in models like SAM and its downstream applications, where the adversary is unaware of the specific fine-tuning task and dataset.

## 3 PRELIMINARIES

### 3.1 FINE TUNING SAM

Let $F_0$ denote SAM, which consists of an image encoder $f_0$, a prompt encoder, and a mask decoder, and let $D_d$ represent the downstream task-specific dataset to which SAM ($F_0$) is adapted. The

objective of the fine-tuning process is to optimize the model parameters so that the downstream SAM model, $F_d$, performs well on the new task. Formally, the fine-tuning process is formulated as:

$$\min_{\Theta_d} \mathbb{E}[\mathcal{L}(F_d(x; \Theta_d), y)], \quad (x, y) \in D_d, \tag{1}$$

where $\Theta_d$ denotes the set of parameters of $F_d$ to be optimized, $\mathcal{L}$ is the task-specific loss function, and $(x, y)$ is an image-mask pair in the fine-tuning dataset. In this work, we consider widely-used PEFT fine-tuning methods such as LoRA (Hu et al., 2022) and Adapter (Wu et al., 2025).

## 3.2 Adversarial Attack Transferability

Transferable adversarial samples are those crafted to fool one model (typically a surrogate model) and can also be used to mislead another model (typically a target model), even if the target model has different parameters or was trained on different data. Given an adversarial example $x' = x + v$ generated on a surrogate model $F_0$, transferability is observed when $x'$ yields high loss on a target model $F_d$, that is,

$$\mathcal{L}(F_d(x'; \Theta_d), y) > \mathcal{L}(F_d(x; \Theta_d), y). \tag{2}$$

High transferability implies that $x'$ generalizes across models in its adversarial effect.

## 4 Problem Statement

Our main problem is how adversarial attacks generated from SAM can be transferred to downstream SAM through fine-tuning. We consider a similar adversarial setting as in Xia et al. (2024): how the adversary constructs transferable adversarial samples from the original SAM without knowing the downstream dataset and accessing the downstream SAM.

**Adversary Setting.** We assume that the adversary has no access to the dataset $D_d$ used during the PEFT fine-tuning of the downstream SAM model. However, the adversary is aware of the fine-tuning structure: the specific PEFT method applied (e.g., LoRA or adapters), as well as the full structural details of the PEFT configuration, including the LoRA rank and the insertion points of the adaptation layers. Additionally, the adversary has full access to the open-source SAM $F_0$, but no direct access to the downstream fine-tuned SAM $F_d$. Given a benign input sample $x$ without knowing the corresponding segmentation mask $y$, the adversary aims to generate a perturbation $v$ such that the adversarial sample $x' = x + v$ causes the downstream SAM $F_d$ to fail in producing a correct segmentation. The adversary's objective can be formally expressed as:

$$x'^{\star} = \arg \max_{x' \in \mathcal{B}(x, \epsilon)} \mathcal{L}(F_d(x'; \Theta_d), y), \tag{3}$$

where $\epsilon$ bounds the perturbation under an $\ell_p$ norm constraint, and $\mathcal{B}(x, \epsilon)$ denotes the $\epsilon$-radius range of the adversarial sample $x'$ centered at $x$ under the $\ell_p$ norm.

This setting is realistic for the following reasons. (1) It is often easier for an adversary to generate adversarial examples using the publicly available SAM rather than the downstream SAM, which may be privately fine-tuned (Xia et al., 2024). (2) Knowing the fine-tuning structure is a reasonable assumption since PEFT techniques may be open-sourced and reused without structural modifications. (3) Given a benign sample $x$, the adversary may not know its correct label $y$ for the downstream SAM (Xia et al., 2024), making the attack more challenging.

**Downstream SAM.** Given the open-source SAM $F_0$ with parameters $\Theta_0$ and a downstream segmentation dataset $D_d$, the downstream model parameters $\Theta_d$ are obtained by:

$$\Theta_d = \arg \min_{\Theta_d} \mathbb{E}_{(x,y) \sim D_d} [\mathcal{L}(F_d(x; \Theta_d), y)]. \tag{4}$$

The fine-tuning process leads to a mapping gap between $F_0$ and $F_d$ as $F_0$ is not trained with the downstream dataset $D_d$. Thus, for two models with the input-output mapping gap, from the adversary's perspective, this gap leads to a transferability problem when generating adversarial samples.

However, it is infeasible to compute the loss $\mathcal{L}(F_d(x; \Theta_d), y)$ directly, since the true label $y$ for the downstream task is unknown to the adversary. Instead, the adversary seeks to generate an adversarial sample $x'$ that maximizes the distance in its encoded feature representation from the original

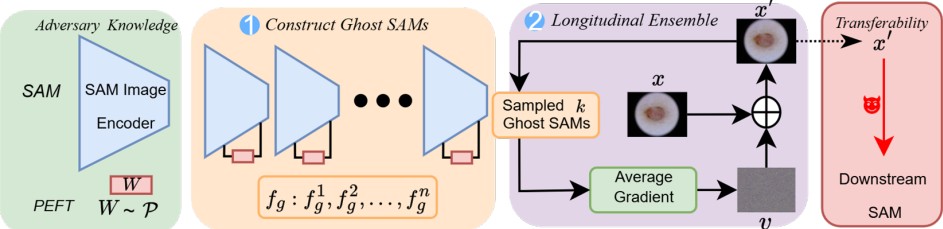

Figure 1: An overview of the attack process of our SETA method. We use LoRA as an example to illustrate our approach, but the method is applicable to other fine-tuning techniques such as adapters.

encoded feature representation in the downstream model. For simplicity, we let $f_0$ and $f_d$ denote the image encoders of the original SAM and the downstream SAM, respectively, and let $f(x; \theta)$ represent the output of an image encoder $f$ for an input $x$ and parameters $\theta$ of $f$.

**Definition 1** (**Optimal Attack on Downstream SAM**). *Given a SAM image encoder $f_0$ and a downstream SAM image encoder $f_d$ with unknown parameters $\theta_d$, we define the optimal adversarial sample $x'^\star$ as the solution to:*

$$x'^\star = \arg\max_{x' \in \mathcal{B}(x,\epsilon)} \|f_d(x'; \theta_d) - f_d(x; \theta_d)\|_2^2, \tag{5}$$

*where $\mathcal{B}(x, \epsilon)$ denotes the $\ell_\infty$-ball of radius $\epsilon$ around $x$. The attack is optimal when it maximally distorts the feature embeddings produced by the downstream encoder $f_d$.*

## 5 METHODOLOGY

We propose the SETA (Structure-Exploiting Transferable Attack) method to generate highly transferable attacks from the original SAM that can successfully cause the downstream SAM to mispredict. The SETA has two steps: (1) construct ghost SAMs; (2) attack generation via longitudinal ensemble.

### 5.1 CONSTRUCTING GHOST SAMS

We adopt the idea of constructing ghost networks in Li et al. (2020b), which proposed to generate additional networks by applying feature-level perturbations (e.g., Dropout Erosion and Skip Connection Erosion). These ghost networks act together as a multi-network ensemble without requiring retraining from scratch, and show strong performance in enhancing attack transferability.

However, this approach is less effective in our problem due to the mapping gap between the original SAM and its downstream fine-tuned variants. As we will demonstrate in Section 6, directly applying feature-level perturbations fails to yield effective transferability in these adversary settings.

Inspired by the concept of ghost networks, we instead construct *ghost SAMs* by leveraging the knowledge of the fine-tuning structure used in the downstream SAM, such as LoRA or adapter modules. This allows us to simulate more accurately the downstream model and improve adversarial transferability without requiring access to its training data and parameters. For example, if the downstream SAM uses LoRA as the fine-tuning method, we construct ghost SAMs by adding the LoRA weight matrix $\Delta W_g = AB$, where $A \in \mathbb{R}^{d \times r}$ and $B \in \mathbb{R}^{r \times k}$.

However, a key challenge in leveraging the fine-tuning structure for ghost network construction is how to determine the values of the added parameters. Since the downstream dataset used for fine-tuning is unknown to the adversary, it is difficult to accurately estimate these parameters. To address this difficulty, we propose to approximate the added parameters by sampling from a Gaussian distribution with a predefined mean and range of standard deviations, $\mathcal{P}$. This approach allows us to simulate multiple potential downstream SAMs by varying the added weights across the ghost networks. As we demonstrate in Section 6, while an accurate estimation of the added parameter distribution yields the highest transferability, even a coarse approximation of the parameter range is sufficient to help achieve competitive attack transferability.

As shown in Figure 1, the adversary constructs a set of ghost SAM image encoders $f_g^1, \ldots, f_g^n$, each derived from the original SAM image encoder $f_0$ by inserting PEFT layers with parameters $\Delta W_g^j \sim \mathcal{P}$. The output deviation between the original SAM and the $j$-th ghost SAM is defined as:

$$f_0(x; \theta_0) - f_g^j(x; \theta_0 + \Delta W_g^j), \quad j = 1, 2, \ldots, n, \tag{6}$$

where $\Delta W_g^j \sim \mathcal{P}$ represents the sampled PEFT parameters for the $j$-th ghost SAM.

## 5.2 ATTACK GENERATION USING LONGITUDINAL ENSEMBLE

We adopt the *longitudinal ensemble* method in Li et al. (2020b) to improve the efficiency of perturbation generation during the adversarial attack process. The longitudinal ensemble strategy leverages a different, randomly constructed ghost SAMs in each attack iteration. Unlike conventional ensemble-based attacks that aggregate outputs or gradients across all ensemble models, the longitudinal ensemble selects one single model or a few sampled models per iteration.

Specifically, given a set of ghost SAMs, $f_g^1, \ldots, f_g^n$, at each iteration, the adversary randomly selects $k$ models, $f_g^{j_1}, \ldots, f_g^{j_k}$, with $j_i \in [1, n]$ for $i = 1, \ldots, k$, to compute the average gradient used for updating the adversarial sample. These selected models are then used in the standard gradient-based attack methods (e.g., MI-FGSM) to iteratively refine the perturbation. By diversifying the update directions over the iterations through randomized model selection, the attack gains improved transferability without incurring the computational cost of aggregating all ghost models at each step.

The procedure of our proposed SETA method is detailed in Algorithm 1. In the first stage (Lines 1–5), SETA constructs a set of ghost SAMs by leveraging prior knowledge of the downstream fine-tuning structure. Each ghost SAM is constructed by injecting sampled parameters into the original SAM model. In the second stage (Lines 8–12), the constructed ghost SAMs are used to generate an adversarial example $x'$. At each of the $T$ attack iterations, $k$ ghost SAMs are randomly selected, the gradients are computed and averaged by using the ghost models, and the adversarial image $x'$ is updated accordingly. The perturbation is then projected into the predefined $\epsilon$-bounded adversarial region to ensure that the modification remains within the acceptable limits.

---

**Algorithm 1** Structure-Exploiting Transferable Attack (SETA)

---

**Require:** An input image $x$, the number of ghost SAMs $k$, parameter distribution $\mathcal{P}$, step size $\alpha$, and the number of maximum iterations $T$
**Ensure:** An adversarial image $x'$
1: **// Construct Ghost SAMs**
2: **for** $j = 1$ to $k$ **do**
3:     Sample LoRA parameters $\Delta W_g^j \sim \mathcal{P}$
4:     Construct ghost SAM parameters $\theta_g^j$ (attach LoRA/adapter modules)
5:     Define ghost SAM $f(x; \theta_g^j)$
6: **end for**
7: Initialize $x' \leftarrow x$
8: **// Perform Longitudinal Ensemble Attack**
9: **for** $t = 1$ to $T$ **do**
10:     Randomly select $f_g^{j_1}, \ldots, f_g^{j_k}$, where $j_i \in [1, n]$
11:     Compute averaged gradient: $\nabla_x L = \frac{1}{k} \sum_{i=1}^{k} \nabla_x \left\| f_g^{j_i}(x'; \theta_g^{j_i}) - f_g^{j_i}(x; \theta_g^{j_i}) \right\|_2^2$
12:     Update the adversarial example: $x' \leftarrow x' + \alpha \cdot \text{sign}(\nabla_x L)$
13:     Project onto $\epsilon$-ball: $x' \leftarrow \text{clip}(x', \mathcal{B}(x, \epsilon))$
14: **end for**
14: **return** $x' = 0$

---

## 5.3 THEORETICAL ANALYSIS

In this subsection, we present a theoretical analysis of how our SETA method provides higher attack transferability using the constructed PEFT layers.

Let the loss function $L(\theta, x')$ measure the discrepancy between the feature embeddings of an adversarial sample $x'$ produced by a surrogate model $f$ (either $f_0$ or a ghost SAM) with parameters $\theta$ and

those produced for the corresponding clean sample $x$, that is,

$$L(\theta, x') = \|f(x';\theta) - f(x;\theta)\|_2^2. \tag{7}$$

The image encoder $f_g$ of a ghost SAM has parameters $\theta_g = \theta_0 + \Delta W_g$, where $\theta_0$ is the parameters of the original SAM image encoder $f_0$ and $\Delta W_g$ is PEFT parameters sampled from a distribution $\mathcal{P}$. Below we formalize the alignment property of the ghost SAMs' input gradients with respect to the downstream SAM model.

**Theorem 1** (**Ghost SAM Advantage for a Typical Downstream Model**). *Let the original, downstream, and ghost image encoders have parameters*

$$\theta_d = \theta_0 + \Delta W_d, \qquad \theta_g = \theta_0 + \Delta W_g,$$

*where $\Delta W_g \sim \mathcal{N}(0, \sigma^2 I)$ and $\Delta W_d$ is a fixed (targeted) downstream SAM fine-tuning parameters.*
*Define*

$$\mathbf{v}_0(x') := f_0(x') - f_0(x), \qquad J_\Delta(x') := J_\theta(x') - J_\theta(x),$$

*with $J_\theta(x) = \frac{\partial f}{\partial \theta}\big|_{\theta=\theta_0}$. Assume:*

*(A1) **Local linearity in parameters**: for all $x' \in \mathcal{B}(x,\epsilon)$ and sufficiently small $\Delta W$,*

$$f(x';\theta_0 + \Delta W) - f(x;\theta_0 + \Delta W) \approx \mathbf{v}_0(x') + J_\Delta(x')\Delta W.$$

*(A2) **Typical downstream fine-tuning**: there exists $\delta \in (0,1)$ such that for all $x' \in \mathcal{B}(x,\epsilon)$,*

$$\left| \Delta W_d^\top J_\Delta(x')^\top J_\Delta(x')\Delta W_d - \sigma^2 \|J_\Delta(x')\|_F^2 \right| \le \delta\,\sigma^2 \|J_\Delta(x')\|_F^2, \tag{8}$$

$$\left| \mathbf{v}_0(x')^\top J_\Delta(x')\Delta W_d \right| \le \delta\,\sigma\,\|\mathbf{v}_0(x')\|\,\|J_\Delta(x')\|_F. \tag{9}$$

*(These bounds hold with high probability when $\Delta W_d \sim \mathcal{N}(0, \sigma^2 I)$ in high dimension.)*

*(A3) **Misalignment / non-degeneracy**: the maximizer of $\|\mathbf{v}_0(x')\|^2$ is not also a maximizer of $\|J_\Delta(x')\|_F^2$ on $\mathcal{B}(x,\epsilon)$.*

*Let*

$$x'_0 := \arg\max_{x' \in \mathcal{B}(x,\epsilon)} \|\mathbf{v}_0(x')\|^2, \qquad x'_g := \arg\max_{x' \in \mathcal{B}(x,\epsilon)} \left( \|\mathbf{v}_0(x')\|^2 + \sigma^2 \|J_\Delta(x')\|_F^2 \right).$$

*Then for the targeted downstream model $f_d$,*

$$\|f_d(x'_g) - f_d(x)\|^2 \ \ge\ \|f_d(x'_0) - f_d(x)\|^2 \ -\ O(\delta),$$

*and the inequality is strict when $\delta$ is sufficiently small under (A3).*

The proof of Theorem 1 is in Appendix A. The theorem states that if the fine-tuning layer in ghost SAM in our SETA satisfies the same Gaussian distribution as the downstream SAM, then our SETA using the ghost SAM serves as a good surrogate for the true downstream loss. As a result, the adversarial example that maximizes the ghost objective produces at least as much damage to the downstream model as the one that only uses the original SAM, up to a small error controlled by $\delta$. Moreover, if the two terms are not maximized at the same point and $\delta$ is small enough, our SETA-generated attack is strictly more damaging.

## 6 EXPERIMENTS

In this section, we evaluate our SETA method on various downstream SAMs and datasets. We first discuss the experimental setting and compare our method with existing transferable attack methods. Then, we show the relation between the attack transferability and the loss distribution near the generated adversarial samples. Finally, we conduct ablation evaluations to examine how the change of parameters influences the results of SETA.

## 6.1 EXPERIMENTAL SETTING

**Downstream SAMs and Datasets.** We conduct experiments on four downstream variants of SAM: **SAMed** (Zhang & Liu, 2023), **Med-SAM** (Wu et al., 2025), **BC-SAM** (Cai et al., 2024), and **TS-SAM** (Yu et al., 2024). **SAMed** fine-tunes the SAM image encoder using LoRA, and jointly optimizes it with the prompt encoder and mask decoder. **Med-SAM** employs a lightweight adaptation strategy for medical image segmentation. **BC-SAM** combines a LoRA-based SAM with a cross-domain autoencoder to improve performance on blood cell image segmentation. **TS-SAM** uses the lightweight Convolutional Side Adapter (CSA) and Multi-scale Refinement Module (MRM) inside the SAM image encoder to capture the downstream image features. We evaluate these downstream SAM models on the synapse multi-organ segmentation dataset (Landman et al., 2018), the CVC-ClinicDB polyp dataset (Bernal et al., 2015), a blood cell test dataset [1], and the COD10K dataset (Fan et al., 2021) for camouflaged object detection, respectively. For evaluation, we adopt the expected IoU metric to assess the performance of Med-SAM and BC-SAM, use the mean dice similarity score (mDSC) metric for SAMed, and structural similarity ($S_\alpha$) for TS-SAM. All our experiments are conducted on NVIDIA A10 GPUs with 23 GB of memory.

**Baselines.** We compare our SETA method with several baseline attack methods: (1) MI-FGSM (Dong et al., 2018), (2) Ghost Net (Li et al., 2020b), (3) PGN (Ge et al., 2023), and (4) MUI-GRAT (Xia et al., 2024).

**Implementation.** We implement our SETA method by sampling $k = 3$ ghost SAMs per iteration, with the total number of ghost SAMs set as $n = k \times T$, where $T$ is the number of attack iterations. The adapter parameters for the ghost SAMs are sampled from a Gaussian distribution with a mean of 0 and a standard deviation range of (0.01, 0.15). In our ablation study below, we investigate how variations in this distribution affect the attack performance. For fair comparison, we use MI-FGSM as the base gradient updating strategy for Ghost Net, MUI-GRAT, and our SETA. We evaluate the attacks using three different perturbation bounds: $\epsilon = 10/255, 15/255, 20/255$ (for simplicity, we use 10,15,20 to represent the $\epsilon$ in below tables). We set the attack step size as $\alpha = 2/255$ and the number of iterations as $T = \epsilon * 255$. For MI-FGSM, PGN, and MUI-GRAT, we adopt the same settings as described in Xia et al. (2024).

## 6.2 RESULTS

We report the main experimental results in Table 1, which show that our proposed SETA method achieves the best attack performance across the various baselines and downstream SAM variants.

Among the baseline methods, UMI-GRAT achieves the best attack performance on SAMed, while PGN performs the best on the other two downstream models. The strong transferability of UMI-GRAT on SAMed is largely due to its injection of gradient noise, which helps mitigate gradient deviation during optimization. This technique has been shown to be critical for improving transferability, as discussed in Xia et al. (2024). However, the heuristic setting of the gradient noise yields diminished performance on the other two models. We argue that mitigating the mapping gap via gradient noise injection depends heavily on the gradient noise estimation, and it is more difficult to attain a precise estimation of the gradients caused by the original SAM and its downstream models in black-box adversarial scenarios. In contrast, our proposed SETA framework, which requires estimation of the PEFT parameter distribution, offers a more feasible approach. This is because the adversary can observe the PEFT parameters from the other fine-tuned models and obtain reasonable approximations of the parameter distribution without needing to access the internal gradients and model parameters.

As shown in Xia et al. (2024), mitigating gradient discrepancies is essential for improving attack transferability among cross-domain models. This finding supports our core hypothesis that SETA significantly enhances transferability by considering the mapping gap between the original SAM and its downstream fine-tuned variants. By constructing PEFT layers to simulate this gap, the generated ghost SAMs effectively capture shared vulnerabilities across the downstream models, therefore improving the robustness and generalization of the adversarial attacks.

---

[1] https://github.com/AnoK3111/BC-SAM/tree/main?tab=readme-ov-file

| Method | SAMed (Synapse) | | | Med-SAM (Polyp) | | | BC-SAM (Blood Cell) | | | TS-SAM (COD) | | |
|---|---|---|---|---|---|---|---|---|---|---|---|---|
| | 10 | 15 | 20 | 10 | 15 | 20 | 10 | 15 | 20 | 10 | 15 | 20 |
| | mDSC ↓ | | | IoU ↓ | | | IoU ↓ | | | $S_\alpha$ ↓ | | |
| No Attack | 79.8 | | | 0.77 | | | 0.96 | | | 0.84 | | |
| MI-FGSM | 65.6 | 63.2 | 61.5 | 0.71 | 0.68 | 0.62 | 0.95 | 0.95 | 0.95 | 0.60 | 0.54 | 0.51 |
| Ghost Net | 65.8 | 64.0 | 61.9 | 0.71 | 0.68 | 0.63 | 0.95 | 0.95 | 0.95 | 0.77 | 0.70 | 0.57 |
| PGN | 55.6 | 40.6 | 22.3 | 0.65 | 0.38 | 0.26 | 0.94 | 0.93 | 0.91 | 0.39 | 0.36 | 0.36 |
| MUI-GRAT | 42.9 | 29.1 | 9.8 | 0.69 | 0.59 | 0.53 | 0.95 | 0.95 | 0.94 | 0.50 | 0.47 | 0.46 |
| SETA | 17.8 | 5.5 | 2.7 | 0.56 | 0.28 | 0.19 | 0.91 | 0.88 | 0.85 | 0.37 | 0.34 | 0.33 |

Table 1: Comparison of attack performance with different attack bounds $\epsilon$ across various downstream SAMs and datasets. Dark gray marks indicate the best attack setting for the corresponding downstream model, and light gray marks represent the second-best setting.

| $\mathcal{P}$ Setting | SAMed (mDSC↓) | Med-SAM (IoU↓) | BC-SAM (IoU↓) | TS-SAM ($S_\alpha$ ↓) |
|---|---|---|---|---|
| $\mathcal{P}_1$: $\mathcal{N}(0, [0.01^2, 0.20^2])$ | 22.5 | 0.63 | 0.92 | 0.38 |
| $\mathcal{P}_2$: $\mathcal{N}(0, [0.03^2, 0.20^2])$ | 24.3 | 0.58 | 0.93 | 0.38 |
| $\mathcal{P}_3$: $\mathcal{N}(0, [0.05^2, 0.20^2])$ | 25.8 | 0.63 | 0.92 | 0.38 |
| $\mathcal{P}_4$: $\mathcal{N}(0, [0.10^2, 0.20^2])$ | 27.2 | 0.65 | 0.93 | 0.38 |
| $\mathcal{P}_5$: $\mathcal{N}(0, [0.15^2, 0.20^2])$ | 27.5 | 0.63 | 0.92 | 0.39 |
| $\mathcal{P}_6$: $\mathcal{N}(0.01, [0.01^2, 0.15^2])$ | 18.9 | 0.61 | 0.92 | 0.38 |
| $\mathcal{P}_7$: $\mathcal{N}(-0.01, [0.01^2, 0.15^2])$ | 20.3 | 0.63 | 0.92 | 0.38 |
| $\mathcal{P}_8$: $\mathcal{N}(0.02, [0.01^2, 0.15^2])$ | 20.7 | 0.61 | 0.92 | 0.37 |
| $\mathcal{P}_9$: $\mathcal{N}(-0.02, [0.01^2, 0.15^2])$ | 18.1 | 0.60 | 0.92 | 0.37 |

Table 2: Attack performance on three downstream SAMs under various Gaussian distribution settings of $\mathcal{P}$ for ghost SAMs.

In addition, note that the Ghost Net baseline adopts a similar adversarial generation strategy as our proposed SETA. While both methods share the motivation of leveraging model variants to enhance transferability, Ghost Net lacks structural awareness of the fine-tuning mechanism. As demonstrated in our results, although dropout-based ghost nets provide modest improvements, they fail to fully capture the parameter variations introduced by downstream fine-tuning. Consequently, they underperform compared to our structure-aware approach.

## 6.3 RELATION BETWEEN TRANSFERABILITY AND LOSS DISTRIBUTION

In this subsection, we examine why our SETA method yields higher gradient alignment with whitebox attack than the baseline methods.

We begin by visualizing the surface of the final loss function (not the distance of feature embeddings) using a randomly selected image from the synapse dataset (Landman et al., 2018). The loss surface is plotted along two random directions in the input space. Previous works such as Ge et al. (2023); Qin et al. (2022) suggested that adversarial examples found in flat local maxima on the surrogate model tend to transfer better to the target model. However, as shown in Figure 2, we observe that although PGN tries to find a flat local maximum on the distance of feature embeddings, it fails to find a local maximum in some cases, as the final loss may not align with the shape of the feature embeddings. This also indicates that a flat local maximum of the feature embeddings on the original SAM does not necessarily correspond to the most effective adversarial sample for downstream SAMs. We found that it is essential to address the mapping gap between the surrogate model and the target model, which is what our SETA does.

Another notable finding is the difference in loss surfaces between the original SAM and its downstream variants. As illustrated in Figure 2, adversarial examples generated on the original SAM often correspond to local optima that also exist in Med-SAM. However, these local optima can trap the adversarial examples in regions with lower loss—typically below 0.07, as seen in Figure 2a and Figure 2b, which limits their transferability. In contrast, the PGN method is able to discover better adversarial examples by locating flatter optima in SAM, resulting in higher loss regions on Med-SAM and thereby improved transferability. Our SETA method further improves upon this, finding the most effective adversarial examples among all the baselines, as shown in Figure 2e.

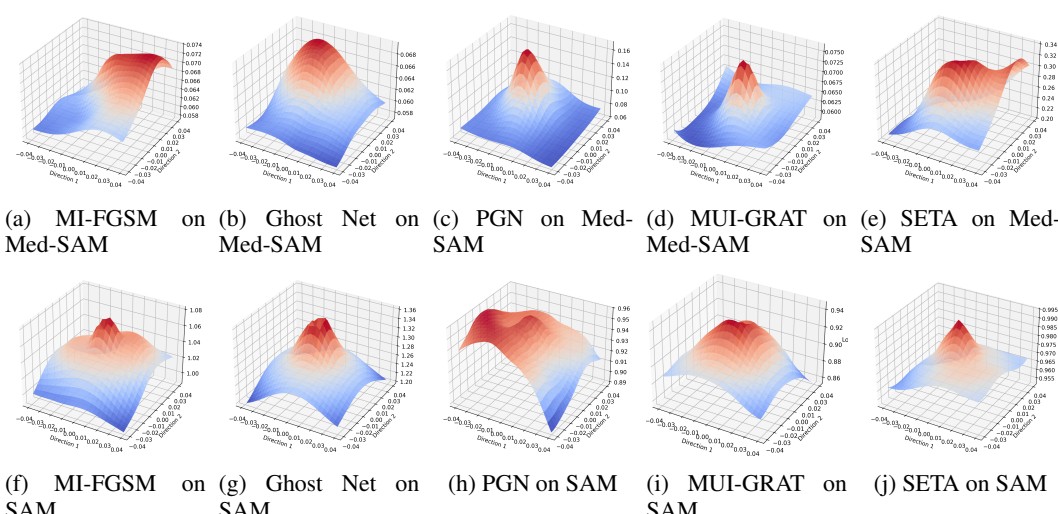

(a) MI-FGSM on Med-SAM
(b) Ghost Net on Med-SAM
(c) PGN on Med-SAM
(d) MUI-GRAT on Med-SAM
(e) SETA on Med-SAM

(f) MI-FGSM on SAM
(g) Ghost Net on SAM
(h) PGN on SAM
(i) MUI-GRAT on SAM
(j) SETA on SAM

Figure 2: Visualization of the loss surface for each attack method on SAM and Med-SAM.

| Distribution Type $\mathcal{P}$ | SAMed (mDSC↓) | Med-SAM (IoU↓) | BC-SAM (IoU↓) | TS-SAM ($S_\alpha$ ↓) |
|---|---|---|---|---|
| Uniform $\mathcal{U}(-a, a)$ | 18.8 | 0.60 | 0.92 | 0.38 |
| Laplace $(0, b)$ | 19.8 | 0.63 | 0.91 | 0.39 |
| Truncated Gaussian | 20.6 | 0.65 | 0.92 | 0.38 |
| Mixture of Gaussians | 18.9 | 0.60 | 0.90 | 0.38 |

Table 3: Impact of different distribution types for ghost SAMs on the attack performance.

## 6.4 ABLATION STUDY

In this subsection, we analyze how the hyperparameters in our SETA framework affect the attack performance. Specifically, we examine the impact of varying the mean and standard deviation in the Gaussian distribution used for sampling ghost SAMs, as well as the influence of different distribution types. We set the perturbation bound to $\epsilon = 10/255$ for all experiments. All other settings follow the implementation details described in Subsection 6.1, with only the analyzed hyperparameters varied.

Table 2 presents the results under different Gaussian distribution settings of $\mathcal{P}$. We observe that SETA achieves the highest attack performance when the sampled PEFT parameter distribution closely matches that of the downstream SAM. For instance, SAMed uses LoRA parameters that follow a Gaussian distribution with a mean near zero and a standard deviation of around 0.1. As a result, the choice of $\mathcal{P}$ influences the attack effectiveness. The impact of standard deviation varies across the three downstream SAMs, suggesting that each model responds differently to parameter variability. In contrast, since the PEFT layers across all three models share a mean close to zero, varying the mean between $-0.02$ and $0.02$ has a minimal effect on the attack performance.

Table 3 reports the impact of different parameter distribution types used to construct ghost SAMs on the attack performance across the three downstream SAM models. For fair comparison, we let all the distribution types lie in the range of $\mathcal{N}(0, [0.01^2, 0.15^2])$. For example, we set the value of $a$ as $a \in [0.0173, 0.2598]$ for the Uniform distribution, and set $b$ as $b \in [0.00707, 0.10607]$ for the Laplace distribution. The detailed parameter settings are given in Appendix D.

## 7 CONCLUSIONS

In this paper, we introduced SETA, a novel structure-exploiting transferable attack framework that targets downstream fine-tuned SAM models without requiring access to their training data and the masks of images. By leveraging knowledge of the fine-tuning structure, SETA constructs ghost SAMs that approximate the downstream SAM models through PEFT layer construction. Our longitudinal ensemble strategy enables efficient adversarial optimization by sampling from a distribu-

tion of structurally aligned ghost SAMs. Extensive experiments across multiple downstream SAM variants and datasets demonstrated that SETA consistently outperforms existing attack baselines in transferability on downstream SAMs. Furthermore, our analyses of loss surfaces and cosine similarity of gradients reveal that structural alignment between the surrogate model and the target model is critical for achieving high transferability. Our findings highlight the importance of exploiting model adaptation structures when designing robust and transferable adversarial attacks.

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

# A    PROOF OF THEOREM

*Proof.* By assumption **(A1)** with $\Delta W = \Delta W_d$, for every $x' \in \mathcal{B}(x, \epsilon)$ we have the local linearization

$$f_d(x') - f_d(x) \approx \mathbf{v}_0(x') + J_\Delta(x')\Delta W_d. \tag{10}$$

Substituting this into the downstream attack loss $L_d(x') := \|f_d(x') - f_d(x)\|^2$ gives

$$L_d(x') \approx \left\|\mathbf{v}_0(x') + J_\Delta(x')\Delta W_d\right\|^2$$
$$= \|\mathbf{v}_0(x')\|^2 + 2\,\mathbf{v}_0(x')^\top J_\Delta(x')\Delta W_d + \Delta W_d^\top J_\Delta(x')^\top J_\Delta(x')\Delta W_d. \tag{11}$$

We define the ghost surrogate objective as the expected linearized downstream loss if we randomly sample fine-tuning parameters from the same Gaussian distribution used to build ghost SAMs.

$$G(x') := \|\mathbf{v}_0(x')\|^2 + \sigma^2\|J_\Delta(x')\|_F^2. \tag{12}$$

Comparing equation 11 and equation 12, we write

$$L_d(x') = G(x') + r(x'), \tag{13}$$

where the residual is

$$r(x') = 2\,\mathbf{v}_0(x')^\top J_\Delta(x')\Delta W_d + \Delta W_d^\top J_\Delta(x')^\top J_\Delta(x')\Delta W_d - \sigma^2\|J_\Delta(x')\|_F^2. \tag{14}$$

By the assumption **(A2)**, for all $x' \in \mathcal{B}(x, \epsilon)$:

$$\left|\Delta W_d^\top J_\Delta(x')^\top J_\Delta(x')\Delta W_d - \sigma^2\|J_\Delta(x')\|_F^2\right| \leq \delta\,\sigma^2\|J_\Delta(x')\|_F^2, \tag{15}$$

$$\left|\mathbf{v}_0(x')^\top J_\Delta(x')\Delta W_d\right| \leq \delta\,\sigma\,\|\mathbf{v}_0(x')\|\,\|J_\Delta(x')\|_F. \tag{16}$$

Applying these bounds to equation 14:

$$|r(x')| \leq 2\left|\mathbf{v}_0(x')^\top J_\Delta(x')\Delta W_d\right| + \left|\Delta W_d^\top J_\Delta(x')^\top J_\Delta(x')\Delta W_d - \sigma^2\|J_\Delta(x')\|_F^2\right|$$
$$\leq 2\delta\,\sigma\,\|\mathbf{v}_0(x')\|\,\|J_\Delta(x')\|_F + \delta\,\sigma^2\|J_\Delta(x')\|_F^2$$
$$=: \varepsilon(x'). \tag{17}$$

Define the uniform bounds

$$M_v := \sup_{x' \in \mathcal{B}(x,\epsilon)} \|\mathbf{v}_0(x')\|, \qquad M_J := \sup_{x' \in \mathcal{B}(x,\epsilon)} \|J_\Delta(x')\|_F. \tag{18}$$

Then from equation 17,

$$\varepsilon(x') \leq 2\delta\,\sigma\,M_v M_J + \delta\,\sigma^2 M_J^2 =: C\,\delta, \tag{19}$$

for some constant $C > 0$ independent of $x'$. Define

$$x'_g := \arg\max_{x' \in \mathcal{B}(x,\epsilon)} G(x'), \qquad x'_0 := \arg\max_{x' \in \mathcal{B}(x,\epsilon)} \|\mathbf{v}_0(x')\|^2. \tag{20}$$

From equation 13 and equation 17 at $x'_g$:

$$L_d(x'_g) = G(x'_g) + r(x'_g) \geq G(x'_g) - \varepsilon(x'_g). \tag{21}$$

Since $x'_g$ maximizes $G$, we have $G(x'_g) \geq G(x'_0)$. Thus:

$$L_d(x'_g) \geq G(x'_0) - \varepsilon(x'_g). \tag{22}$$

Similarly, at $x'_0$:

$$L_d(x'_0) = G(x'_0) + r(x'_0) \leq G(x'_0) + \varepsilon(x'_0), \tag{23}$$

which gives

$$G(x'_0) \geq L_d(x'_0) - \varepsilon(x'_0). \tag{24}$$

Substituting equation 24 into equation 22:

$$L_d(x'_g) \geq L_d(x'_0) - \varepsilon(x'_0) - \varepsilon(x'_g). \tag{25}$$

Using equation 19, we have $\varepsilon(x'_0) + \varepsilon(x'_g) \leq 2C\delta$, so

$$L_d(x'_g) \geq L_d(x'_0) - O(\delta). \tag{26}$$

Under assumption **(A3)**, the maximizers of $\|\mathbf{v}_0(x')\|^2$ and $\|J_\Delta(x')\|_F^2$ do not coincide. Therefore,

$$\gamma := G(x'_g) - G(x'_0) > 0. \tag{27}$$

From equation 13,

$$
\begin{aligned}
L_d(x'_g) - L_d(x'_0) &= \big(G(x'_g) - G(x'_0)\big) + \big(r(x'_g) - r(x'_0)\big) \\
&\geq \gamma - |r(x'_g)| - |r(x'_0)| \\
&\geq \gamma - \varepsilon(x'_g) - \varepsilon(x'_0) \\
&\geq \gamma - 2C\delta.
\end{aligned}
\tag{28}
$$

Thus, if $\delta < \gamma/(2C)$, then

$$L_d(x'_g) > L_d(x'_0). \tag{29}$$

In summary, for general $\delta$ we have equation 26, and under **(A3)** with sufficiently small $\delta$, the strict inequality equation 29 holds. $\square$

## B  ADDITIONAL ABLATION STUDY

| # **of** $k$ | **SAMed** (mDSC↓) | **Med-SAM** (IoU↓) | **BC-SAM** (IoU↓) | **TS-SAM** ($S_\alpha$ ↓) |
|---|---|---|---|---|
| $k = 1$ | 27.3 | 0.58 | 0.93 | 0.37 |
| $k = 2$ | 20.6 | 0.57 | 0.92 | 0.37 |
| $k = 3$ | 17.8 | 0.56 | 0.91 | 0.37 |
| $k = 4$ | 15.5 | 0.56 | 0.91 | 0.36 |
| $k = 5$ | 15.5 | 0.55 | 0.92 | 0.36 |

Table 4: Impact of varying the number of ghost SAMs $k$ for the SETA method on the attack performance.

| **Method** | **Time (s)** on **SAMed** |
|---|---|
| SETA ($k = 1$) | 0.55 |
| SETA ($k = 2$) | 1.12 |
| SETA ($k = 3$) | 1.62 |
| SETA ($k = 4$) | 2.17 |
| SETA ($k = 5$) | 2.70 |
| MI-FGSM | 0.11 |
| Ghost Networks | 1.85 |
| PGN | 1.12 |
| MUI-GRAT | 0.11 |

Table 5: The computational overhead on **SAMed**.

In Table 4, we demonstrate how the number of ghost SAMs used in each attack iteration affects the attack performance of our SETA method. Increasing the number of ghost SAMs may enhance

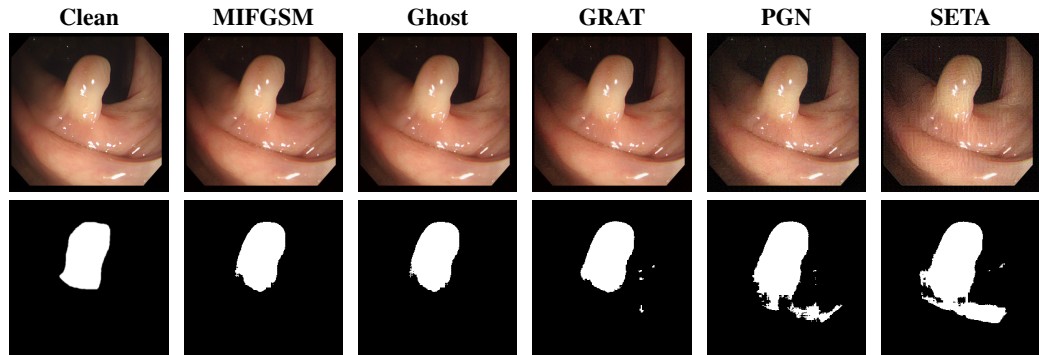

Figure 3: Comparison of different attack methods on the polyp dataset. The top row shows the input/perturbation images; the bottom row shows the predicted masks (or ground truth).

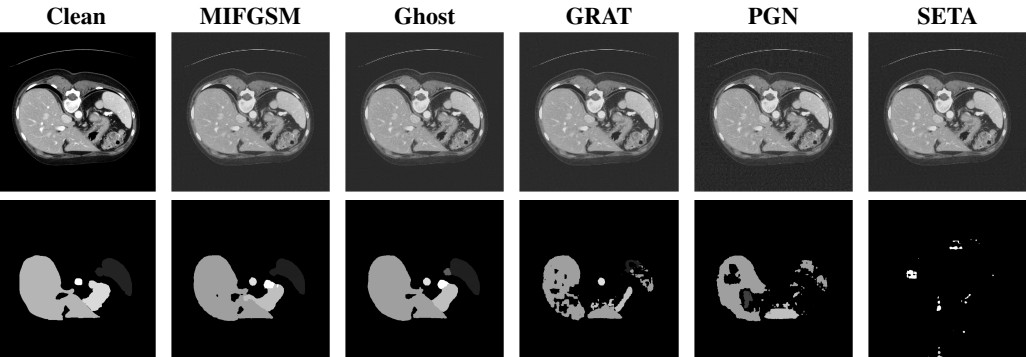

Figure 4: Comparison of different attack methods on Synapse dataset. The top row shows the input/perturbation images; the bottom row shows the predicted masks (or ground truth).

the transferability of the adversarial examples, similar to how using multiple models during attack generation can improve transferability in standard adversarial settings.

We also report the computational overhead for each attack iteration of our SETA method, compared to other baseline approaches. For SETA, we vary the number of ghost SAMs used, since each ghost SAM requires both a forward pass and backpropagation during each attack iteration. As shown in Table 5, the computational overhead increases approximately linearly with the number of ghost SAMs. In contrast, methods like MI-FGSM and MUI-GRAT require only a single gradient update, resulting in significantly lower computational cost. Our SETA method, by leveraging multiple ghost SAMs, results in higher computation time. All timing results are measured on 512×512 input images using a NVIDIA A10 GPU and the SAMed model based on SAM-ViT-B.

Our SETA requires the adversary to know the fine-tuning structure of the downstream SAM. To demonstrate that SETA can also enhance attack transferability when the adversary has only partial knowledge of the structure, we report the attack performance in Table 6 under the same attack setting as Table 1, with attack bound $\epsilon = 10/255$. The performance is evaluated when different numbers of fine-tuning blocks are correctly placed, while the remaining blocks are omitted from the downstream SAM. The correctly placed blocks are sequentially positioned from the input side of the image encoder toward the output side. The results in Table 6 show that the more fine-tuning blocks are correctly placed in the SETA ghost SAM, the better attack performance SETA achieves.

In Table 7, we compare the attack performance of SETA with and without the longitudinal ensemble. The longitudinal ensemble is designed to reduce computational overhead rather than to improve attack transferability. Without the longitudinal ensemble, we aggregate the gradients from all ghost SAMs at every iteration to generate adversarial examples. Although this yields slightly better attack

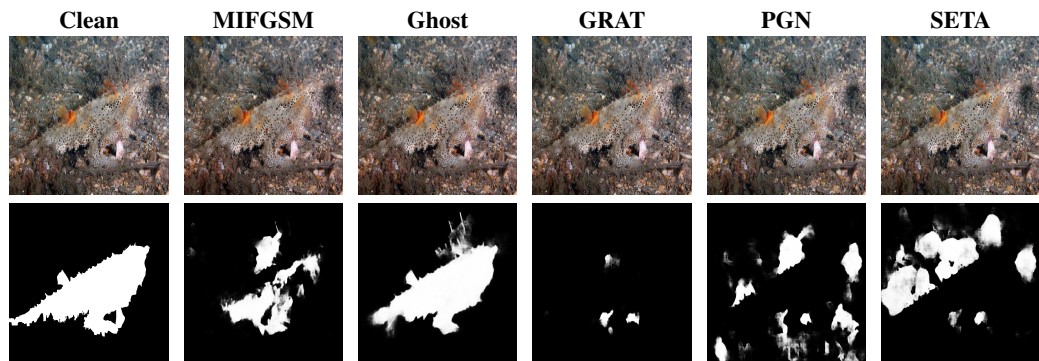

Figure 5: Comparison of different attack methods on COD10K dataset. The top row shows the input/perturbation images; the bottom row shows the predicted masks (or ground truth).

| Model | 2 Blocks | 4 Blocks | 6 Blocks | 8 Blocks | 10 Blocks |
|---|---|---|---|---|---|
| SAMed (mDSC↓) | 66.99 | 38.18 | 25.32 | 22.30 | 19.84 |
| Med-SAM (IoU↓) | 0.325 | 0.294 | 0.293 | 0.291 | 0.289 |
| BC-SAM (IoU↓) | 0.962 | 0.917 | 0.916 | 0.916 | 0.915 |
| TS-SAM ($S_\alpha$↓) | 0.813 | 0.623 | 0.421 | 0.382 | 0.362 |

Table 6: Performance under different numbers of correctly placed blocks.

performance, it requires roughly $\frac{n}{k}$ times more computation, where $n$ is the total number of ghost SAMs and $k$ is the ghost SAMs sampled every iteration in the longitudinal ensemble.

We also implement our attacks and baselines on the AdapterShadow (Jie & Zhang, 2023). Table 9 compares the BER (Balance Error Rate) of different adversarial attacks on AdapterShadow under increasing perturbation budgets $\epsilon$. A higher BER indicates a stronger attack, where

$$\text{BER} = \left(1 - 0.5(\text{TP}/N_p + \text{TN}/N_n)\right) \times 100,$$

with $\text{TP}/N_p$ and $\text{TN}/N_n$ denoting the true positive rates for shadow and non-shadow pixels, respectively. Across all budgets, SETA consistently achieves the highest BER, demonstrating the strongest attack effectiveness among the compared methods.

| $\epsilon * 255$ | SAMed (mDSC↓) | Med-SAM (IoU↓) | BC-SAM (IoU↓) | TS-SAM ($S_\alpha$ ↓) |
|---|---|---|---|---|
| 10 | 17.8/9.4 | 0.28/0.24 | 0.91/0.91 | 0.37/0.35 |
| 15 | 5.5/2.8 | 0.56/0.46 | 0.88/0.88 | 0.34/0.33 |
| 20 | 2.7/1.6 | 0.19/0.12 | 0.85/0.85 | 0.33/0.31 |

Table 7: The attack performance of our SETA w/o longitudinal ensemble.

| Method | Med-SAM (Polyp) IoU ↓ | | | TS-SAM (COD) $S_\alpha$ ↓ | | |
|---|---|---|---|---|---|---|
| | 10 | 15 | 20 | 10 | 15 | 20 |
| No Attack | | 0.82 | | | 0.89 | |
| MI-FGSM | 0.72 | 0.68 | 0.63 | 0.62 | 0.55 | 0.52 |
| Ghost Net | 0.69 | 0.66 | 0.59 | 0.80 | 0.75 | 0.61 |
| PGN | 0.66 | 0.45 | 0.29 | 0.47 | 0.39 | 0.36 |
| MUI-GRAT | 0.67 | 0.58 | 0.52 | 0.53 | 0.47 | 0.46 |
| SETA | 0.60 | 0.34 | 0.24 | 0.44 | 0.40 | 0.36 |

Table 8: Attack performance on Med-SAM (ViT-H) and TS-SAM (ViT-H) under different perturbation budgets $\epsilon$.

| $\epsilon * 255$ | MI-FGSM (BER↑) | Ghost Net (BER↑) | PGN (BER↑) | MUI-GRAT (BER↑) | SETA (BER↑) |
|---|---|---|---|---|---|
| 10 | 0.64 | 0.63 | 0.68 | 0.67 | 0.69 |
| 15 | 0.64 | 0.66 | 0.70 | 0.71 | 0.81 |
| 20 | 0.65 | 0.73 | 0.83 | 0.85 | 1.38 |

Table 9: Attack performance on **AdapterShadow** across different attack methods. BER is reported with different perturbation budgets $\epsilon$ with an average of 0.61 BER when there is no attack. The higher the BER, the better the attack performance.

## C  VISUALIZATION OF THE GENERATED ADVERSARIAL IMAGES AND PREDICTIONS

We show the generated adversarial images produced by different attack methods under a fixed perturbation bound, along with their corresponding predicted masks. The visualizations include randomly selected examples from the Synapse and Polyp datasets. As illustrated in Figure 3 and Figure 5, the results are consistent with the quantitative findings in Table 1, confirming that our SETA method achieves the most effective attack performance with its generated perturbation images.

## D  DETAILS OF HYPARAMETERS

Table 10 provides the paramter settings for Table 3. The distributions of the ghost SAMs impact the attack performance of our SETA.

| Distribution Type and Description | Parameter Value(s) |
|---|---|
| Uniform $\mathcal{U}(-a, a)$ | $a = 0.0173$ |
| Laplace $\mathcal{L}(0, b)$ | $b \in [0.0071, 0.1061]$ |
| Truncated Gaussian $\mathcal{N}(0, \sigma^2)$ truncated at $\pm 2\sigma$ | $\sigma \in [0.01, 0.15]$ |
| Mixture of Gaussians: $0.5 \cdot \mathcal{N}(\mu_1, \sigma_1^2) + 0.5 \cdot \mathcal{N}(\mu_2, \sigma_2^2)$ | $(\mu_1, \sigma_1) = (-0.1, 0.05)$ $(\mu_2, \sigma_2) = (0.1, 0.05)$ |

Table 10: Parameter settings used in Table 3.

## E  DISCUSSION

**Limitations.** Our SETA approach assumes full knowledge of the fine-tuning structure, such as the type of PEFT method and its insertion locations. In practice, this information may not always be available to the adversary, particularly in strict black-box scenarios. Nevertheless, our experiments demonstrate that even with only partial knowledge of the fine-tuning structure, the adversary can still achieve relatively strong attack performance. Another limitation arises when the target model

| Main idea | SETA | Ghost Net | MI-FGSM / PGN | MUI-GRAT |
|---|---|---|---|---|
| Ensemble multiple SAM variants | ✓ | ✓ | ✗ | ✗ |
| Meta-initialization and gradient noise | ✗ | ✗ | ✗ | ✓ |
| Stable gradient direction | ✗ | ✗ | ✓ | ✗ |

Table 11: Comparison of main ideas used by different transferable attacks.

is fine-tuned in a substantially different manner (e.g., full fine-tuning or domain-specific prompt tuning), in which case the attack effectiveness cannot be guaranteed. Furthermore, this work does not explore possible defense strategies against SETA.

**Possible defense solutions.** Our work focuses on developing highly transferable adversarial attacks and does not systematically evaluate how SETA behaves under existing defense mechanisms. We briefly discuss several possible defenses. First, adversarial training (Bai et al., 2021; Li et al., 2024a) augments the training data with adversarial examples and optimizes the model; this strategy has been widely explored in the literature and is expected to substantially reduce the effectiveness of SETA if applied to SAM or its downstream variants. Second, input-transformation defenses apply a pre-processing step to the input (e.g., denoising, JPEG compression, or other transformations) before feeding it to the model. The impact of such transformations on the robustness of semantic segmentation models (though not SAM specifically) has been investigated in prior work (Liu et al., 2020b; Arnab et al., 2018), and similar ideas could be adapted to our setting. Finally, adversarial example detection methods (Alotaibi & Rassam, 2023; Maag et al., 2024; Zhang et al., 2025), which aim to detect suspicious inputs before inference, provide a defense solution that could be combined with the above strategies. A thorough evaluation of these defense solutions against SETA is an important direction for future work.

**Discussion of major novelty.** Our core contribution is leveraging the structure of downstream SAM to enhance adversarial transferability. We observe that fine-tuning layers in downstream SAM exhibit near-zero mean and low variance (e.g., 0.05–0.1). By emulating this distribution, we reduce the gradient mismatch between SAM and its downstream variants. Unlike (Xia et al., 2024), which adds fixed Gaussian noise to feature losses, our method avoids estimating gradient ranges, leading to more precise and effective attacks. We provide a detailed comparison table 11 that summarizes the differences between our work and the baselines used in our experiments.

# F  THE USE OF LARGE LANGUAGE MODELS (LLMS)

We use LLMs to assist with writing tasks, including grammar checking and improving readability.

