# OpenReview forum: "Exploiting Fine-Tuning Structures to Improve Adversarial Transferability on Downstream SAM"
_ICLR.cc/2026/Conference — Submitted to ICLR 2026_

### Official Review · Reviewer_CHBq · 2025-10-27

**Soundness:** 2
**Presentation:** 3
**Contribution:** 3
**Rating:** 6
**Confidence:** 3

**Summary:**

This paper presents the adversarial attack method SETA for downstream fine-tuning of the SAM model. The authors‘ approach is ingenious. By leveraging fine-tuning structures to construct "ghost models" and combining with a longitudinal integration strategy, the attack's transferability is enhanced. The paper has a clear structure and a well-defined problem definition, making it easy for readers to understand. However, the experiments were only conducted under similar fine-tuning scales and did not verify their effectiveness in more complex or defensive conditions. Overall, this research idea is novel and provides valuable inspiration for structure-aware adversarial attacks.

**Strengths:**

This paper is written clearly and concisely with a rigorous structure. The author provides an exact problem definition and symbol explanation at the beginning, enabling readers to quickly grasp the main research framework and hypotheses, which greatly enhances the clarity and logical coherence of the paper. Furthermore, the proposed SETA method is conceptually innovative. It ingeniously utilizes the structural characteristics of downstream fine-tuning to construct a "ghost SAM" and combines a longitudinal integration strategy to improve transferability.

**Weaknesses:**

Although the SETA method proposed in this paper is innovative and has achieved good experimental results, it still has some shortcomings. Firstly, the method assumes that the downstream LoRA/Adapter parameters follow a Gaussian distribution, but there is a lack of empirical verification. The rationality of this assumption under different tasks remains uncertain. Secondly, SETA relies on the known fine-tuning structure and is applicable to PEFT models, but its generalization to full-parameter or prompt fine-tuning scenarios is limited. The experimental part mainly focuses on similar fine-tuning scales and does not analyze the performance under different fine-tuning intensities, structural differences, or defense mechanisms. Moreover, the independent contributions of the Ghost SAM and the longitudinal integration parts have not been evaluated separately, and the source of performance improvement is not clear. Overall, the main limitations of the paper lie in insufficient verification of theoretical assumptions, narrow experimental scope, and insufficient robustness analysis.

**Questions:**

1. The author assumes that the parameters of LoRA or Adapter in the downstream SAM model can be approximately represented by a Gaussian distribution, and based on this, "Ghost SAM" was constructed in theoretical analysis and experiments. Although this assumption makes the method simpler and the modeling more convenient, the paper does not provide empirical evidence to prove whether this Gaussian distribution assumption is truly valid. In fact, different tasks, LoRA configurations, and optimization settings will lead to different distributions of these parameters, so the universality of this assumption in different fine-tuning scenarios remains uncertain.
2. The paper introduces a longitudinal ensemble strategy to enhance the stability of adversarial attacks. However, the experiments do not include an independent comparison between using only Ghost SAMs without the ensemble and using only the ensemble strategy without constructing Ghost SAMs. As a result, it isn't easy to attribute the performance gains to either component clearly. This makes the contribution of the longitudinal ensemble module somewhat ambiguous and leaves the overall effect of each part insufficiently validated.
3. The paper mainly evaluates the attack performance in terms of success rate and performance degradation, but does not investigate the robustness of the proposed attack under input perturbations or common defense mechanisms such as adversarial training.
4. The experiments only cover downstream models with similar LoRA/Adapter configurations. It is not clear whether SETA is still effective when the fine-tuning intensity is stronger or the structure is different, which raises questions about its generalization ability in different fine-tuning scales.

---

> ### Author Response · Authors · 2025-11-26
> **Rebuttal by Authors**
>
> ##### Dear reviewer,
>
> We sincerely appreciate the reviewer’s careful reading and insightful comments.
>
> #### Question 1- provide empirical evidence to prove whether this Gaussian distribution
>
> We agree with the reviewer that the parameter distribution of LoRA/Adapter fine-tuning can vary under different tasks, configurations, and optimization settings.
>
> In our experiments, we manually checked the fine-tuning parameters of all downstream models we used and observed that they are mostly well-fitted by a Gaussian distribution at the PEFT layers.
>
> | Model   | Best-fit dist. | #Layers | Avg. mean | Avg. std |
> |---------|----------------|---------|-----------|----------|
> | TS-SAM  | Gaussian       | 309     | 0.22400   | 0.08488  |
> | TS-SAM  | Laplace        | 192     | 0.13000   | 0.03082  |
> | TS-SAM  | Uniform        | 33      | -0.01140  | 0.12060  |
> | SAMed   | Gaussian       | 48      | 0.00019   | 0.09221  |
> | BC-SAM  | Gaussian       | 36      | -0.00004  | 0.03128  |
> | BC-SAM  | Laplace        | 12      | -0.00003  | 0.00659  |
> | Med-SAM | Gaussian       | 79      | 0.00010   | 0.03293  |
> | Med-SAM | Uniform        | 65      | -0.05335  | 0.10648  |
>
> This is the reason why we adopt Gaussian perturbations in our Ghost SAM construction in the experimental section. Moreover, although different settings may lead to uncertainty, we have conducted several experiments under different fine-tuning settings, and the results consistently show that our method remains valid across these variations.
>
> #### Question 2- each part insufficiently validated
>
> We thank the reviewer for pointing out the need to clarify the contribution of the longitudinal ensemble strategy. The longitudinal ensemble strategy does not improve the attack performance; instead, it is efficient to use all the ensemble models. Opposite to the longitudinal ensemble strategy, another strategy is to use all the gradients from all the ensemble models every attack iteration, which will be slow. We add an ablation experiment that applies SETA with/without the longitudinal ensemble, so that the improvement brought by the ensemble module can be evaluated independently. The table below indicates that the longitudinal ensemble does not contribute to the attack performance. Although without the longitudinal ensemble yields slightly better attack performance, it requires roughly $\frac{n}{k}$ times more computational time, where $n$ is the total number of ghost SAMs and $k$ is the ghost SAMs sampled every iteration in the longitudinal ensemble.
>
> | ε * 255 | SAMed (mDSC↓) | Med-SAM (IoU↓) | BC-SAM (IoU↓) | TS-SAM (Sα↓) |
> |---------|---------------|----------------|---------------|--------------|
> | 10      | 17.8/9.4      | 0.28/0.24      | 0.91/0.91     | 0.37/0.35    |
> | 15      | 5.5/2.8       | 0.56/0.46      | 0.88/0.88     | 0.34/0.33    |
> | 20      | 2.7/1.6       | 0.19/0.12      | 0.85/0.85     | 0.33/0.31    |
>
> #### Question 3.- does not investigate under input perturbations or common defense
>
> We appreciate the reviewer’s concern about robustness under defenses.
>
> This paper focuses on adversarial attacks in terms of transferability, and we do not investigate robustness under input perturbations or common defense mechanisms in experiments. We will add more discussion on potential defense mechanisms against our proposed attacks in the discussion section. We admit that adversarial training is an effective defense and can reduce the effectiveness of our attack. For future work, we will justify the issue.
>
> #### Question 4- whether SETA is still effective when the fine-tuning intensity is stronger or the structure is different
>
> We agree that the fine-tuning scale is an important factor.
>
> We admit that our method is designed for PEFT-based downstream models (e.g., LoRA and adapters), and it cannot be directly applied to full fine-tuning (fine-tuning on all the parameters). Within this scope, our experiments include TS-SAM in Section 6, which has a much stronger fine-tuning intensity than other downstream SAM variants. Our method is still effective on TS-SAM illustrates that it can generalize to relatively stronger PEFT fine-tuning settings. We also show that even though the structures used in the SETA attack are different from the downstream model, our method can still have comparable performance, as shown in Table 6. We will emphasize this limitation and the supporting evidence more clearly in the revision.
>
> ##### We appreciate the reviewer’s comments. Please let us know if further clarification would be helpful.

---

### Official Review · Reviewer_kWj2 · 2025-10-30

**Soundness:** 3
**Presentation:** 3
**Contribution:** 2
**Rating:** 4
**Confidence:** 4

**Summary:**

This paper studies black-box adversarial transferability from the original Segment Anything Model (SAM) to its downstream fine-tuned variants under a parameter-efficient fine-tuning (PEFT) setting. The authors propose SETA, which (1) constructs “ghost” SAMs by injecting sampled PEFT layers (e.g., LoRA or adapters) into the base encoder, and (2) performs a longitudinal ensemble attack by randomly selecting a subset of these ghost models at each iteration. Empirical evaluation against three downstream SAM variants shows that SETA outperforms MI-FGSM, Ghost Networks, PGN, and MUI-GRAT in terms of feature-distance loss.

**Strengths:**

Strengths:

1. The paper is well-structured and easy to follow.

2. A comprehensive empirical study demonstrates the effectiveness of the proposed method.

**Weaknesses:**

1. The attack presumes exact access to the downstream PEFT configuration (insertion points, ranks, etc.), which may not hold in realistic black-box settings where fine-tuning details are proprietary or obfuscated.

2. The selection of ghost parameters $\Delta W$is based on random initialization (sampled from a Gaussian); there is no theoretical analysis or guidance on choosing better parameters.

3. All experiments use variants of the ViT-based SAM. The transferability of SETA across different vision transformers (ViT-H or L) remains unexplored.

4. Adapters are also a common way of fine-tuning SAM, whereas the methodology in this paper focuses on LoRA design and lacks a discussion of adapters.

5. Given that adversarial attacks and defenses are two closely related areas, it would be desirable to add a discussion of adversarial defenses to the relevant work section.

**Questions:**

My major concerns are weaknesses 2, 3, 4, and 5. I would like to raise my score if the author can solve those concerns during the rebuttal.

---

> ### Author Response · Authors · 2025-11-26
> **Rebuttal by Authors**
>
> ##### Dear reviewer,
>
> ##### We sincerely appreciate the insightful comments.
>
> #### Weaknesses 1-attack may not hold in a black-box setting
>
> ##### We acknowledge that SETA relies on knowledge of the specific fine-tuning structure (e.g., LoRA/adapters) to construct the attack, which may not always be realistic and is indeed a major limitation of our approach. However, this fine-tuning structure is often available in practice when SAM is adapted to downstream tasks by reusing publicly released fine-tuning repos or open-sourced checkpoints, without changing the underlying fine-tuning architecture. In such cases, an adversary can reasonably know the fine-tuning structure, and our SETA method can then be applied to achieve highly transferable attacks across downstream SAM variants.
>
> #### Weaknesses 2- guidance on choosing better parameters
>
> ##### We admit that we do not have a theoretical analysis or guidance on choosing those parameters. The reason why we choose Gaussian parameters is that we manually check the parameter distribution of the used downstream models. And these downstream SAMs have the Gaussian distribution in their majority fine-tuning layers.
>
> | Model   | Best-fit dist. | #Layers | Avg. mean | Avg. std |
> |---------|----------------|---------|-----------|----------|
> | TS-SAM  | Gaussian       | 309     | 0.22400   | 0.08488  |
> | TS-SAM  | Laplace        | 192     | 0.13000   | 0.03082  |
> | TS-SAM  | Uniform        | 33      | -0.01140  | 0.12060  |
> | SAMed   | Gaussian       | 48      | 0.00019   | 0.09221  |
> | BC-SAM  | Gaussian       | 36      | -0.00004  | 0.03128  |
> | BC-SAM  | Laplace        | 12      | -0.00003  | 0.00659  |
> | Med-SAM | Gaussian       | 79      | 0.00010   | 0.03293  |
> | Med-SAM | Uniform        | 65      | -0.05335  | 0.10648  |
>
> #### Weaknesses 3- experiments only on ViT-based SAM
>
> ##### We thank the reviewer for the concern about the single version of transformers used in our experiments. We have added an experiment on ViT-H SAM to illustrate the results of our proposed methods on TS-SAM and Med-SAM.
>
> | Method    | Med-SAM (Polyp) IoU↓ ε=10 | Med-SAM IoU↓ ε=15 | Med-SAM IoU↓ ε=20 | TS-SAM (COD) Sα↓ ε=10 | TS-SAM Sα↓ ε=15 | TS-SAM Sα↓ ε=20 |
> |-----------|---------------------------|-------------------|-------------------|-----------------------|-----------------|-----------------|
> | No Attack | 0.82                      | 0.82              | 0.82              | 0.89                  | 0.89            | 0.89            |
> | MI-FGSM   | 0.72                      | 0.68              | 0.63              | 0.62                  | 0.55            | 0.52            |
> | Ghost Net | 0.69                      | 0.66              | 0.59              | 0.80                  | 0.75            | 0.61            |
> | PGN       | 0.66                      | 0.45              | 0.29              | 0.47                  | 0.39            | 0.36            |
> | MUI-GRAT  | 0.67                      | 0.58              | 0.52              | 0.53                  | 0.47            | 0.46            |
> | SETA      | 0.60                      | 0.34              | 0.24              | 0.44                  | 0.40            | 0.36            |
>
> #### Weaknesses 4- lacks a discussion of adapters
>
> ##### We thank the reviewer for suggesting a discussion of the adapter case, since our paper currently uses LoRA mainly as a running example to illustrate the methodology. We acknowledge that LoRA and adapters differ in how ghost SAMs are constructed. For LoRA, the effect of fine-tuning can be written in an additive form, e.g., $f_g^j(x;\theta_g^j) = f_0(x; \theta_0) + \Delta W_g^j(x)$, because the LoRA updates enter the layer transformation linearly. For adapter-based fine-tuning, adapters introduce additional non-linear modules rather than a simple linear offset, as LoRA. Our SETA can adopt both linear and non-linear modules. In the revision, we will clarify this distinction and update Algorithm 1 and the text to use a more general notation in which ghost SAMs are obtained by attaching fine-tuning parameters $ΔW_g^j$ (LoRA or adapters) to the frozen base SAM. Our experiments already include both LoRA-based and adapter-based downstream models, and empirically demonstrate that SETA is effective in both settings.
>
> #### Weaknesses 5- discussion of adversarial defenses
>
> ##### We thank the reviewer for pointing out the lack of discussion on adversarial defenses. In the revised version, we have expanded the discussion section to include relevant adversarial defense strategies for segmentation models and SAM-style architectures, and to position SETA with respect to potential defenses (e.g., robust training).
>
> ##### We appreciate the reviewer’s comments. Please let us know if further clarification would be helpful.

---

> > ### Comment · Reviewer_kWj2 · 2025-11-27
> > **Response to authors**
> >
> > Thanks to the author's detailed response and additional experiments. My major concerns are solved. I thus raise my score to 6.

---

> > > ### Author Response · Authors · 2025-11-27
> > > **Response to reviewer kWj2**
> > >
> > > Dear reviewer,
> > >
> > > Thank you so much for your constructive feedback, which is very helpful and makes our paper stronger!
> > >
> > > Best,
> > >
> > > The Authors

---

### Official Review · Reviewer_of7z · 2025-10-30

**Soundness:** 2
**Presentation:** 3
**Contribution:** 2
**Rating:** 2
**Confidence:** 4

**Summary:**

The paper proposes a method to generate adversarial inputs to the open weight SAM segmentation model in such a way so that the examples transfer also to fine-tuned versions of SAM, without having information about the dataset used for fine-tuning, or the fine-tuned weights. The main idea is based on two pillars: first, a set of "ghost" models is created that contains "randomly fine-tuned" models (meaning that the fine-tuning method is used (eg lora) but the actual perturbations made are random (as opposed to being learned). This set of ghost models is then attacked using a longitudinal ensemble attack (attacking with an iterative method but using only a small subset of the models in each iteration). The method is demonstrated to work well empirically, and some theoretical results are also presented in its support.

**Strengths:**

The paper proposes a method that can be demonstrated to provide favorable results compared to related work.

The targeted problem is interesting, if adversarial examples transfer to fine-tuned variants of foundation models then open weight models represent a vulnerability in this sense.

**Weaknesses:**

The novelty of the paper is rather minimal in terms of the approach itself. Both the idea of using ghost models and longitudinal attacks are known from the literature, this approach is applied in the fine-tuning setup.

The presentation of the paper, especially the formal parts (equation) have many issues, starting with eq (1) which is a very unusual (in fact, incorrect) way of defining the learning problem (should be a minimization of the sum of sample losses or the expectation of the loss), eq (2) is both meaningless and wrong (there is no requirement of "much larger" in general), eq (4) is announced as a parameter update, which it is not (it is the definition of the optimization problem, extended with an initialization statement (that is not needed), definition 1 is quite strange (in eq (5) the "L=" is not needed, also, in the previous sentence x' is defined, but then used as a variable in the eq, eq (6) uses the notation $\Delta W$  which previously was used to denote the changes in the parameters, and immediately used in (6) as the change in the output,

Theorem 1 seems not to be true, and the proof contains basic issues that make the bounding incorrect. The theorem is formulated in such a way that it should hold for any loss $L$, but essentially only the linear case is discussed, as the first step is introducing a linear approximation. The approximation of the expectation of the cosine function is not justified because the numerator and denominator are not independent. But the theorem is quite clearly not true even for the linear case, simply due to symmetry: in the linear case the gradient of the ghost set cannot be closer to that of *every* possible fine-tuning than to that of the original model, while the claim supposed to hold for every $f_d$.

The method works, but for an entirely different, in fact, opposite reason. The random ghost set makes sure that the attack focuses on those directions that lead to adversarial inputs *independently* of the fine-tuning. In other words, the ghost models help *ignore* the fine tuning bits and help focus on the core bits: those directions that are common in the original and fine-tuned model, hence the transferability.

**Questions:**

Can you please revise the theoretical discussion (or remove it completely if that is not possible)?

---

> ### Author Response · Authors · 2025-11-26
> **Rebuttal by Authors**
>
> Dear reviewer,
>
> We thank the reviewer for pointing out the mistakes in the formal part; we sincerely apologize for these errors. We have carefully revised the manuscript accordingly. The main corrections are as follows:
>
> In eq (1), we add the expectation notation to the loss, so that the objective explicitly minimizes the expected total loss.
> In eq (2), we change the notation from ''much larger'' to ''larger''. to avoid an unnecessarily strong statement.
> In eq (4), we revise Eq. (4) by changing it from an update process to a clear optimization definition and removing the parameter initialization term to match the formulation.
> In eq (5), we remove the redundant L to make the equation clearer and avoid confusion.
> In eq (6), we revise the definition to $f_0(x;\theta_0) - f_g^j(x; \theta_0 + \Delta W_g^j), \quad j=1,2,\ldots,n.$ to make the notation explicit and easier to follow.
>
> We thank the reviewer for the detailed critique of Theorem 1 and its proof. Initially, we want to show that our method can work to improve the attack transferability. We apologize again for making mistakes in the previous proof of the theorem. In the revised manuscript, we have rewritten both the statement of Theorem 1 and its proof to address these concerns.
>
> In the revised version, we no longer claim that ghost gradients are “closer” than the original gradient to every possible downstream fine-tuning. Instead, we prove that:
>
> We consider the locally linearized downstream model around the base SAM parameters $\theta_0$: $f(x';\theta_0+\Delta W) - f(x;\theta_0+\Delta W)\approx \mathbf v_0(x') + J_\Delta(x') \Delta W,$ where $\mathbf v_0(x')$ is the base-model difference and $J_\Delta(x')$ is the Jacobian w.r.t. the fine-tuning layer parameters $\Delta W$.
>
> For a downstream fine-tuning $\Delta W_d$, we show that the downstream attack loss
>
> $L_d(x') := |f_d(x') - f_d(x)|^2$
>
> is well approximated by the ghost surrogate object
>
> $G(x') := |\mathbf v_0(x')|^2 + \sigma^2 |J_\Delta(x')|_F^2,$
>
> so that
>
> $L_d(x') = G(x') + O(\delta)\quad \text{for all } x' \in \mathcal B(x,\epsilon).$
>
> Let
>
> $x'_0 := \arg\max _{x' \in \mathcal B(x,\epsilon)} |\mathbf v_0(x')|^2, \qquad x'_g := \arg\max _{x' \in \mathcal B(x,\epsilon)} G(x').$ The theorem then shows
>
> $|f_d(x'_g) - f_d(x)|^2 \ge |f_d(x'_0) - f_d(x)|^2 - O(\delta),$
>
> and, under a non-degeneracy assumption, this inequality is strict when $\delta$ is sufficiently small.
>
> Maximizing the ghost objective $G$ yields perturbations that approximately maximize the downstream loss and can strictly outperform perturbations that only optimize the base-model term $|\mathbf v_0(x')|^2$, for downstream fine-tunings.
>
> The new proof is built around three explicit assumptions:
>
> (A1) Local linearity in parameters.
> We assume that for all $x' \in \mathcal B(x,\epsilon)$ and small $\Delta W$, $f(x';\theta_0+\Delta W) - f(x;\theta_0+\Delta W)\approx \mathbf v_0(x') + J_\Delta(x') \Delta W.$ This makes that the analysis is local and based on a first-order expansion in $\Delta W$.
> It allows us to write $L_d(x')$ as a quadratic form in $\Delta W_d$ involving $\mathbf v_0(x')$ and $J_\Delta(x')$.
>
> (A2)  Downstream fine-tuning.
> We assume that the actual downstream fine-tuning parameters $\Delta W_d$ satisfy a Gaussian distribution with variance $\sigma^2$. We require that:
>
> The quadratic term $\Delta W_d^\top J_\Delta ^\top J_\Delta \Delta W_d$ is close (up to factor $\delta$) to its Gaussian expectation $\sigma ^2 |J_\Delta |_F^2$.
>
> The cross term  $\mathbf v_0^\top J_\Delta \Delta W_d$ is small (again controlled by $\delta$).
>
> Under this assumption, we can show that
>
> $L_d(x') = G(x') + r(x'), \quad |r(x')| \le \varepsilon(x') = O(\delta),$
>
> for all $x'$ in the perturbation ball.
>
> (A3) Misalignment / non-degeneracy.
> We assume that the maximizer of $|\mathbf v_0(x')|^2$ is not also a maximizer of $|J_\Delta(x')|_F^2$.
> This means that there is some trade-off between these two terms, so the combined objective
>
> $G(x') = |\mathbf v_0(x')|^2 + \sigma^2 |J_\Delta(x')|_F^2$
>
> attains its maximum at a point (x'_g) that is generally different from $x'_0$.
> Under this misalignment, we have $G(x'_g) > G(x'_0)$, and when $\delta$ is small the positive gap in $G$ dominates the $O(\delta)$ residual error, yielding $L_d(x'_g) > L_d(x'_0).$
>
> Intuitively, the  theorem formalizes the following idea:
>
> Ghost models sample fine-tuning directions around the base SAM according to a Gaussian distribution.
> In the locally linear regime, the downstream loss for fine-tuning is well approximated by a ghost surrogate $G(x')$ that depends only on base-model quantities and the Jacobian norm.
> By maximizing $G(x')$, SETA finds perturbations that approximately maximize the downstream loss and, under mild misalignment, strictly outperform attacks that only optimize the original SAM.
>
> We hope this clarification makes the revised theoretical result and its limitations clear and addresses the reviewer’s concerns about the original Theorem 1.

---

### Official Review · Reviewer_7gzd · 2025-10-30

**Soundness:** 3
**Presentation:** 2
**Contribution:** 2
**Rating:** 6
**Confidence:** 3

**Summary:**

This paper investigates the adversarial transferability between SAM and its downstream fine-tuned variants. The authors introduce SETA, a method that exploits knowledge of the fine-tuning structures of downstream models to enhance transferability. Specifically, SETA constructs ghost SAMs that emulate possible fine-tuned variants by injecting sampled PEFT parameters, and employs a longitudinal ensemble strategy to iteratively craft more transferable perturbations. Experimental results demonstrate that SETA consistently outperforms established baselines such as MI-FGSM, PGN, GhostNet, and MUI-GRAT. Moreover, theoretical analysis and loss-surface visualizations substantiate the claim that structural awareness fosters better gradient alignment, thereby improving cross-model transferability.

**Strengths:**

1. This paper is among the first to take a practical perspective by explicitly modeling downstream fine-tuning structures to improve the adversarial transferability of SAM in domain-specific scenarios.

2. The idea of constructing "ghost SAMs" through sampling PEFT parameters is both novel and insightful. It extends the ghost network concept in a structure-aware manner that effectively captures downstream adaptation mechanisms.

3. Extensive experiments across four distinct SAM variants consistently demonstrate substantial improvements in attack performance.

4. The paper is well-organized and thorough, featuring detailed algorithmic pseudocode, comprehensive ablation studies (e.g., distribution types, hyperparameters, and number of ghost SAMs), and thoughtful discussions on limitations.

**Weaknesses:**

1. While the paper claims to operate within the same context as [1], it lacks a thorough comparison with that baseline and does not clearly articulate the distinct problem formulation or contribution beyond the prior work.

2. Table 4 indicates that SETA outperforms competing methods even when (k = 1). However, the paper does not examine whether SETA would maintain this advantage if other methods were also permitted to ensemble multiple open-source models, which could affect the fairness of the comparison.

3. The motivation for adopting Gaussian sampling could be further clarified, as Table 3 suggests that different parameter distribution types lead to comparable performance.

4. Most downstream models evaluated in the experiments belong to the medical domain, and the improvement observed on COD is relatively marginal. A deeper analysis of this phenomenon would strengthen the paper. Additionally, including more diverse downstream models (e.g., shadow segmentation or other non-medical tasks) would help validate the generality of the proposed approach.

[1] Song Xia, Wenhan Yang, Yi Yu, Xun Lin, Henghui Ding, Lingyu Duan, and Xudong Jiang. Transferable adversarial attacks on SAM and its downstream models. arXiv preprint arXiv:2410.20197, 2024.

**Questions:**

Please answer the questions proposed in the weakness section.

**Details Of Ethics Concerns:**

The proposed method can be used to attack other models, potentially posing a security threat to real-world applications that rely on them.

---

> ### Author Response · Authors · 2025-11-26
> **Rebuttal by Authors**
>
> Dear reviewer,
>
> We thank the reviewer for the constructive feedback, which helped us improve the paper.
>
> #### Weakness 1 — Comparison and contribution clarity.
>
> We acknowledge that our setting is related to [1] in problem level, we both focus on generating highly transferable attack on downstream SAM. Our key difference is on the methodology level, that we exploit the fine-tuning structure of downstream SAM models to enhance transferability, without meta-initialization or gradient noise as in [1]. We also benchmark against strong transferable attacks and find that our method consistently outperforms them. To strengthen the comparison, we have added ViT-H SAM experiments.
>
> [1] Song Xia, Wenhan Yang, Yi Yu, Xun Lin, Henghui Ding, Lingyu Duan, and Xudong Jiang. Transferable adversarial attacks on SAM and its downstream models. arXiv preprint arXiv:2410.20197, 2024
>
> #### Weakness 2 — Fairness of comparisons.
>
> We agree that fairness is important, and we note that SETA indeed leverages an ensemble of multiple ghost SAMs. We would like to clarify the problem setting: our goal is to generate effective perturbations on _downstream_ SAM models in a transfer-based setting where the adversary can only access the original SAM and has no knowledge of the downstream task or dataset. In this setting, baselines such as MI-FGSM, PGN, and MUI-GRAT, in their standard form, compute gradients only on the original SAM. By contrast, the Ghost Net baseline also constructs ghost networks by modifying the original SAM, which is conceptually similar to our approach. Importantly, our method does not ensemble multiple independently trained models; we only ensemble ghost SAMs that share the same base SAM and differ by a small parameter change in their fine-tuning layers. This keeps the comparison fair while allowing us to exploit the fine-tuning structure of SAM for better transferability.
>
> #### Weakness 3 — Choice of Gaussian sampling.
>
> We appreciate the constructive feedback from the reviewer. We chose Gaussian sampling after empirically inspecting the parameter distributions of multiple downstream models. For each fine-tuning layer, we fit three candidate distributions—Gaussian, Laplace, and Uniform—and select the best fit based on standard goodness-of-fit tests. We observe that the majority of layers are well-approximated by Gaussians, which directly motivates our choice of a Gaussian prior for sampling. A detailed statistical summary of the fine-tuning layer parameter distributions is reported below.
>
> | Model   | Best-fit dist. | #Layers | Avg. mean | Avg. std |
> |---------|----------------|---------|-----------|----------|
> | TS-SAM  | Gaussian       | 309     | 0.22400   | 0.08488  |
> | TS-SAM  | Laplace        | 192     | 0.13000   | 0.03082  |
> | TS-SAM  | Uniform        | 33      | -0.01140  | 0.12060  |
> | SAMed   | Gaussian       | 48      | 0.00019   | 0.09221  |
> | BC-SAM  | Gaussian       | 36      | -0.00004  | 0.03128  |
> | BC-SAM  | Laplace        | 12      | -0.00003  | 0.00659  |
> | Med-SAM | Gaussian       | 79      | 0.00010   | 0.03293  |
> | Med-SAM | Uniform        | 65      | -0.05335  | 0.10648  |
>
> #### Weakness 4 — Generality and backbone/optimizer choices.
>
> We thank the reviewer for pointing out the problem.
>
> The reviewer notes that the performance gains on COD are marginal. We believe this is largely due to the characteristics of the COD dataset, where the foreground object and background share highly similar visual patterns. In such a setting, even small perturbations can dramatically disrupt the segmentation quality for all methods. As a result, under our chosen perturbation budgets $\epsilon \in {10, 15, 20}$, the relative improvement of our method over strong baselines such as PGN appears marginal, even though the absolute degradation in performance is still significant.
>
> To further demonstrate the generality of SETA, we additionally evaluate it on a shadow-segmentation downstream SAM [1]. We report the Balanced Error Rate (BER) as the attack metric, where a higher value indicates stronger attack performance. The results show that SETA consistently achieves higher BER than the baselines, indicating that our approach generalizes well across both medical and non-medical segmentation tasks.
>
> | ε * 255 | MI-FGSM (BER↑) | Ghost Net (BER↑) | PGN (BER↑) | MUI-GTAT (BER↑) | SETA (BER↑) |
> |---------|----------------|------------------|------------|-----------------|-------------|
> | 10      | 0.64           | 0.63             | 0.68       | 0.67            | 0.69        |
> | 15      | 0.64           | 0.66             | 0.70       | 0.71            | 0.81        |
> | 20      | 0.65           | 0.73             | 0.83       | 0.85            | 1.38        |
>
> [1] Jie, Leiping, and Hui Zhang. "AdapterShadow: Adapting segment anything model for shadow detection." arXiv preprint arXiv:2311.08891 (2023).
>
> We appreciate the reviewer’s insightful comments and will incorporate these revisions.

---

> > ### Comment · Reviewer_7gzd · 2025-11-28
> >
> > Thanks to the authors for the feedback.
> >
> > Regarding Weakness 1, I suggest providing a more detailed comparison table that clearly summarizes the differences between this work and other closely related approaches.
> >
> > For Weakness 2, do the authors refer to ensembling ghost networks? If so, would it be possible to use them as ensemble baselines for a fairer comparison. For example, by computing gradients not only on the original SAM in baselines? This will highlight the advantages of ensembling ghost SAMs.
> >
> > For Weakness 3, the meaning of "Avg. mean" and "std" is unclear. A clearer explanation or a more intuitive way to illustrate the degree of best fit would help readers better understand the results.
> >
> > For Weakness 4, should "MUI-GTAT" actually be "MUI-GRAT"?

---

> > > ### Author Response · Authors · 2025-11-29
> > > **Rebuttal by Authors**
> > >
> > > Thanks to the reviewer for the further constructive suggestions for improving our paper.
> > >
> > > 1. Suggest a table for comparison
> > >
> > > We appreciate the suggestion from the reviewer. We have modified our manuscript in the discussion section
> > > in the appendix to provide a table for comparing this work with other related approaches. We believe that this table can
> > > clarify our main differences and contributions to other related work.
> > >
> > > | Main idea                              | SETA          | Ghost Net    | MI-FGSMand PGN | MUI-GRAT      |
> > > |----------------------------------------|---------------|--------------|----------------|---------------|
> > > | Ensemble Multiple SAM variants         | $\checkmark $ | $\checkmark$ | $\times$       | $\times$      |
> > > | Meta-initialization and Gradient Noise | $\times$      | $\times$     | $\times$       | $\checkmark $ |
> > > | stable gradient direction              | $\times$      | $\times$     | $\checkmark$   | $\times$      |
> > >
> > >
> > >
> > > 2. ensembling with ghost networks
> > >
> > > We thank the reviewer for pointing out this important experiment comparison with using baselines to ensemble ghost networks.
> > > We want to first clarify that those methods like MUI-GRAT, PGN, and MI-FGSM can aggregate with the ghost networks, and in our original
> > > experiment settings, our baseline ghost network is using MI-FGSM, as we clarify in " For fair comparison, we use MI-FGSM
> > > as the base gradient updating strategy for Ghost Net, MUI-GRAT, and our SETA," in section 6.1. So the original baseline Ghost Net
> > > is already implemented as "ghost net + MI-FGSM". Below, we add new experiments to show how PGN and MI-FGSM work with ghost net.
> > > And these experiments show the advantage of ensembling ghost SAMs.
> > >
> > > <table>
> > >   <thead>
> > >     <tr>
> > >       <th rowspan="3">Method</th>
> > >       <th colspan="3">SAMed (Synapse)</th>
> > >       <th colspan="3">Med-SAM (Polyp)</th>
> > >       <th colspan="3">BC-SAM (Blood Cell)</th>
> > >       <th colspan="3">TS-SAM (COD)</th>
> > >     </tr>
> > >     <tr>
> > >       <th>10</th><th>15</th><th>20</th>
> > >       <th>10</th><th>15</th><th>20</th>
> > >       <th>10</th><th>15</th><th>20</th>
> > >       <th>10</th><th>15</th><th>20</th>
> > >     </tr>
> > >     <tr>
> > >       <th colspan="3">mDSC &darr;</th>
> > >       <th colspan="3">IoU &darr;</th>
> > >       <th colspan="3">IoU &darr;</th>
> > >       <th colspan="3">S<sub>&alpha;</sub> &darr;</th>
> > >     </tr>
> > >   </thead>
> > >   <tbody>
> > >     <tr>
> > >       <td>PGN W/O Ghost Net</td>
> > >       <td>56.3/55.6</td><td>46.6/40.6</td><td>38.6/22.3</td>
> > >       <td>0.69/0.65</td><td>0.61/0.38</td><td>0.45/0.26</td>
> > >       <td>0.94/0.94</td><td>0.93/0.93</td><td>0.92/0.91</td>
> > >       <td>0.60/0.39</td><td>0.49/0.36</td><td>0.41/0.36</td>
> > >     </tr>
> > >     <tr>
> > >       <td>MUI-GRAT W/O Ghost Net</td>
> > >       <td>47.2/42.9</td><td>44.3/29.1</td><td>31.5/9.8</td>
> > >       <td>0.74/0.69</td><td>0.72/0.59</td><td>0.71/0.53</td>
> > >       <td>0.95/0.95</td><td>0.95/0.95</td><td>0.94/0.94</td>
> > >       <td>0.80/0.50</td><td>0.76/0.47</td><td>0.72/0.46</td>
> > >     </tr>
> > >   </tbody>
> > > </table>
> > >
> > > From this table, we observe that combining Ghost Net with PGN or MUI-GRAT does **not** improve attack transferability.
> > > This is consistent with the behavior of the original MI-FGSM and Ghost Net: MI-FGSM alone outperforms Ghost Net,
> > > And in our implementation, Ghost Net is essentially Ghost Net + MI-FGSM.
> > >
> > >
> > >
> > >
> > > 3. Explanation to illustrate the "Avg. mean" and "std".
> > >
> > > We thank the reviewer for pointing out the unclear terminology. For the first table in the previous response, **“Avg. mean”** and **“Std”** describe the distribution of the **fine-tuning parameters** used in the downstream SAM.
> > >
> > > For a given downstream SAM, let $\Delta W_d$ denote the parameter updates of all fine-tuned layers. We first compute statistics **layer by layer**:
> > >
> > > - For each fine-tuned layer $\ell$, we compute the mean $\mu_{d,\ell}$ and standard deviation $\sigma_{d,\ell}$ of all parameter values in $\Delta W_{d,\ell}$.
> > >
> > > We then aggregate these per-layer statistics:
> > >
> > > - **“Avg. mean”** is the average of the per-layer means $\mu_{d,\ell}$ over all fine-tuned layers.
> > > - **“Std”** is the average of the per-layer standard deviations $\sigma_{d,\ell}$ over all fine-tuned layers.
> > >
> > > In other words, we first obtain a mean and standard deviation for each fine-tuning layer separately, and then compute the final reported values by averaging these layer-wise quantities.
> > >
> > >
> > > 4. Typo
> > > Thanks for pointing this out. We apologize for the typo.
> > >
> > > We would like to again thank the reviewer for their constructive and insightful feedback.

---

### Author Response · Authors · 2025-12-02
**Author Rebuttal by Authors**

We sincerely appreciate all the constructive feedback from the reviewers. We summarize our revisions as follows:

1. Ghost parameter selection and Gaussian distribution (Reviewers 7gzd, kWj2, CHBq).
We have clarified the selection process of ghost parameters and added our empirical observations to justify why we chose the Gaussian distribution in our initial setting.

2. Methodology-level comparison and new AdapterShadow SAM experiment (Reviewer 7gzd).
We have strengthened the methodology-level comparison between our method and the baselines, highlighting the key conceptual differences from related works. In addition, we added new experimental results on AdapterShadow SAM to demonstrate the generalization ability of our method on both medical and non-medical tasks.

3. Writing issues and theorem revision (Reviewer of7z).
We have carefully addressed all the writing issues pointed out by the reviewer and revised our theorem and proof.

4. Defense discussion, adapter attack, and ViT-H experiments (Reviewer kWj2).
We have added further discussion on potential defense strategies and attacks on adapter-based models. We also included new experiments on ViT-H SAM. We sincerely thank Reviewer kWj2 for raising the score after these revisions.

5. SETA ablation and scope clarification (Reviewer CHBq).
We have added new experiments evaluating our SETA method without the longitudinal ensemble strategy. We also clarified the scope of our method on Adapter and LoRA-based models and emphasized that our focus is on adversarial attacks.

We would like to thank all the reviewers again for their insightful comments and valuable suggestions, which have significantly helped us improve the paper.

---

### Meta-Review · Area_Chair_Nxd1 · 2026-01-05

**Summary:**

The paper proposes black-box, transferable adversarial attacks for the Segment Anything (SAM) model. The authors study the setting where SAM has been finetuned (using PEFT) for downstream tasks, and the authors assume knowledge of this. The authors (1) construct "ghost" SAMs by randomly sampling PEFT layers into the base adapter and (2) randomly selecting a subset of these ghost models at each iteration, to ensure that the adversarial attack transfers across these different settings. The authors show empirically that the proposed method outperforms existing works in terms across a range of datasets.

Reviewers generally appreciated the motivation for the work, and its empirical results. Most concerns raised by reviewers during the rebuttal, relating to experiments, were addressed well.

However, Reviewer of7z raised serious concerns about the correctness of the method, finding errors in the proposed theorem, and errors in the notation and description of the method. This also suggests that the insights posited by the authors about the method cannot be trusted either.

Although the empircal results of the proposed method are strong, the AC does not support accepting a paper with mathematical errors in it. And it will also require another round of review to ensure that these have been corrected. Authors are therefore encouraged to incorporate all of the reviewers' feedback and resubmit to another venue.

**Reviewer Concerns:**

Reviewer 7gzd: Concerns addressed
Reviewer of7z: Concerns about theoretical correctness not addressed
Reviewer kWj2: Concerns addressed
Reviewer CHBq. Concerns addressed. However, review appears to be LLM generated

**Reviewer Scores:**

Reviewer 7gzd: Remain at weak accept
Reviewer of7z: Remain at reject
Reviewer kWj2: Improve to weak accept
Reviewer CHBq. Remain at reject. However, review appears to be LLM generated

---

### Decision · Program_Chairs · 2026-01-26

Reject